# Semaphorin 3A-Neuropilin-1 Signaling Modulates MMP13 Expression in Human Osteoarthritic Chondrocytes

**DOI:** 10.3390/ijms232214180

**Published:** 2022-11-16

**Authors:** Sabine Stöckl, Johanna Reichart, Magdalena Zborilova, Brian Johnstone, Susanne Grässel

**Affiliations:** 1Department of Orthopaedic Surgery, Experimental Orthopaedics, Centre for Medical Biotechnology (ZMB), University of Regensburg, 93053 Regensburg, Germany; 2Department of Orthopaedics and Rehabilitation, Oregon Health & Science University, Portland, OR 97239, USA

**Keywords:** osteoarthritis, semaphorin-3A, neuropilin-1, chondrocyte metabolism, MMP13, AKT signaling

## Abstract

Osteoarthritis (OA) is a complex disorder of diarthrodial joints caused by multiple risk factors and is characterized by articular cartilage destruction as well as changes in other articular tissues. Semaphorin 3A (Sema3A), known to be a chemo-repellent for sensory nerve fibers, has recently been implicated in cartilage OA pathophysiology. We demonstrated that the expression of SEMA3A and its receptor neuropilin-1 (NRP1) are synchronously upregulated in chondrocytes isolated from knee cartilage of OA patients compared to non-OA control chondrocytes. In addition, we observed that during in vitro passaging of OA chondrocytes, the Nrp-1 level increases, whereas the Sema3A level decreases. In this study, we aimed to uncover how Sema3A-Nrp-1 signaling affects metabolism and viability of OA chondrocytes via siRNA-mediated inhibition of Nrp-1 expression. We observed a decreased proliferation rate and an increase in adhesion and senescence after Nrp-1 silencing. Moreover, MMP13 gene expression was reduced by approximately 75% in NRP1 knockdown OA chondrocytes, whereas MMP13 expression was induced by Sema3A treatment in control (nt siRNA) OA chondrocytes, accompanied by an impaired AKT phosphorylation. These findings suggest a potential catabolic function of Sema3A signaling in OA chondrocytes by inducing MMP13 expression and by compromising pro-survival AKT activation. We propose that targeting the Sema3A-Nrp-1 signaling axis might be an opportunity to interfere with OA pathogenesis and progression.

## 1. Introduction

Osteoarthritis (OA) is a degenerative disease of the whole joint and is a major cause of disability, especially in the elderly population. Currently, there is no treatment capable of altering or halting its progression; although great attention has been paid to unraveling the molecular mechanisms underlying OA pathogenesis, they still remain elusive. The major radiographic characteristics of late-stage OA include osteophyte formation, changes in subchondral bone resulting in sclerosis and above all the progressive loss of the articular cartilage [1,2]. Unlike other tissues, cartilage does not contain blood vessels and is not deeply innervated, indicating that it is a hostile environment for the spreading of nerve fibers. The presence of nerve repellent factors may be one cause of this lack of innervation of healthy cartilage, which changes during OA pathogenesis [3]. Sensory neuropeptides like substance P (SP) and α Calcitonin Gene-Related Peptide (αCGRP), released from ingrowing and/or existing sensory nerve fibers, can affect chondrocyte metabolism and thus influence the progression of OA [4].

Semaphorins are a class of repellents, best known for their role in guiding axons to their targets in the developing nervous system by providing repulsive signals [5,6,7]. Semaphorin 3A (Sema3A), a selective repellent of SP- and αCGRP-positive sensory nerve fibers, is expressed in developing cartilage and bone. Gomez and colleagues showed that Sema3A and its receptor neuropilin (nrp)1 are expressed in resting, pre-hypertrophic, and hypertrophic chondrocytes in growth cartilage before the onset of neurovascular invasion during endochondral ossification [8]. A murine model with Sema3A mutations confirmed that Sema3A signaling participates in cartilage development by demonstrating cartilage abnormalities [9]. Sema3A and Nrp1 are overexpressed in human OA chondrocytes in comparison with healthy articular chondrocytes [10]. Hence, the level of Sema3A and Nrp1 in the joint may affect pain sensation arising in the joints and may accelerate the beginning or progressing of OA by directly affecting chondrocyte metabolism and vitality. In our study, we compared the gene and protein expression of Sema3A and Nrp1 in OA and non-OA chondrocytes and analyzed the influence of chondrocyte aging/senescence (by in vitro passaging) on Sema3A and Nrp1 expression level. To gain more details on the impact of the Sema3A-Nrp1-signaling axis on the pathobiology of OA chondrocytes, we introduced a transient *Nrp1* knockdown in OA chondrocytes, followed by functional assays and gene expression analysis. Finally, we stimulated OA chondrocytes with recombinant Sema3A, followed by signaling pathway analysis.

## 2. Results

### 2.1. Gene and Protein Expression of Sema3A and Nrp1 in OA Chondrocytes Compared with non-OA Chondrocytes

We compared the expression of Sema3A and Nrp1 in OA chondrocytes and non-OA chondrocytes on gene and protein level. The gene expression analysis showed a significant increase in *SEMA3A* (4,5-fold) and of *NRP1* (6,3-fold) mRNA (Figure 1a), whereas on the protein level, only Sema3A displayed a significantly higher expression in the OA group compared to the non-OA group (18%). The protein level of Nrp1 revealed no significant difference in Western blot analysis using whole cell lysates for analysis of both proteins (Figure 1b,c).

### 2.2. Effects of In Vitro Passaging on Nrp1 and Sema3A Expression in OA Chondrocytes

Chondrocytes lose their chondrogenic phenotype during aging/senescence, a scenario that can be recapitulated by in vitro expansion (passaging). Thereby, the chondrocytes exhibit a more fibroblast-like morphology characterized by decreased collagen type II expression and increased collagen type I expression. The so-called dedifferentiation process of chondrocytes leads to a loss of their original functional characteristics, needed for tissue reconstruction and/or regeneration [11]. We aimed to understand whether passaging of OA chondrocytes influenced the gene and protein expression levels of Sema3A and Nrp1, to identify a potential link of aging and dedifferentiation processes. *NRP1* gene expression increased significantly (~25%) in passage 3, compared with passage 0, 1 and 2 (Figure 2a). In contrast, we measured a ~50% reduction on the *SEMA3A* gene expression level in comparison with passage 0 already after the first passage. After passage 2 and 3, the *SEMA3A* gene expression level further decreased to ~25% compared to the respective passage 0 cells (Figure 2b). Western blot analysis confirmed the increase in Nrp1 (Figure 2c) and the decrease in Sema3A on the protein level (Figure 2d). Figure 2e shows a representative Western blot image for Sema3A (~82 kDa), Nrp1 (~120 kDa) and β-actin (~40 kDa) protein expression in chondrocytes from OA patients expanded in monolayer until passage 3.

### 2.3. Nrp1 Knockdown Affects Proliferation, Adhesion and Senescence in OA-Chondrocytes

To better understand the role of Sema3A-Nrp-1 signaling in OA chondrocyte metabolism and vitality, we transiently inhibited the expression of Nrp1 with specific siRNA probes. A reproducible knockdown of 80–90% was generated in RNA and protein levels (Figure 3a,b). Subsequently, we analyzed whether OA chondrocytes display changes in cell biology and viability after Nrp1 silencing. We examined proliferation, adhesion, senescence, apoptosis and migration. The proliferation of *NRP1* knockdown cells was significantly decreased in comparison with control (non-target; nt siRNA) cells (Figure 3c). The adhesion rate increased in OA chondrocytes after *NRP1* knockdown (Figure 3d), and so did the activity of the senescence-associated marker SA-β-galactosidase (Figure 3e) after Nrp1 inhibition. Apoptosis and migration were not altered in *NRP1*-knockdown chondrocytes, shown in Appendix A.

### 2.4. NRP1 Knockdown Affects MMP13 and BCL-2 Expression

We analyzed a panel of different genes in OA chondrocytes after *NRP1* knockdown to test whether typical markers for adhesion (cell-substrate interaction), for chondrogenic (de-) differentiation, or for proliferation/apoptosis are affected. The gene expression of matrix metalloproteinase (*MMP*) 13—a protease that degrades cartilage—was significantly down-regulated in *NRP1* knockdown cells compared with control (nt siRNA) cells (reduction of about 75%), whereas the expression of *BCL-2*, a pro-survival and anti-proliferative protein, was slightly increased (increase of about 25%) (Figure 4a). Even though we considered the increase in *BCL-2* expression to be statistically significant, it was not biologically relevant, as the difference to the control cells was less than two-fold. MMP13 is critical for OA progression and degenerative processes in joints, thus, we analyzed the Mmp13 protein level after *NRP1* knockdown in OA chondrocytes from four different patients. Although the gene expression of *MMP13* was decreased in all analyzed samples after *NRP1* knockdown, we detected in two out of the four patients’ samples a strong decrease in pro-Mmp13 protein expression (patient 1 and 3) in comparison with the control (nt siRNA) cells (Figure 4b), indicating a very patient-specific influence of Nrp1 on Mmp13 protein expression.

### 2.5. Sema3A-Mediated Induction of MMP13 Gene Expression Can Be Blocked via NRP1 Knockdown

To establish that the decrease in *MMP13* gene expression after *NRP1* knockdown is directly connected to the disturbed Nrp1-Sema3A signaling axis, we stimulated *NRP1* knockdown cells and control cells (nt siRNA) with three different concentrations (1, 10, 100 ng/mL) of recombinant Sema3A and analyzed subsequently the *MMP13* gene expression level (Figure 5). *NRP1* knockdown cells showed no alteration of *MMP13* gene expression after treatment with 1 or 10 ng/mL Sema3A, but a decrease after stimulation with 100 ng/mL Sema3A (Figure 5a) was measured. The same Sema3A treatment regimen (100 ng/mL) induced a significant increase in *MMP13* gene expression in control cells (nt siRNA) in comparison with non-Sema3A-treated control cells (nt siRNA) (Figure 5b). Hence, the Sema3A-mediated induction of *MMP13* gene expression can be counteracted via *NRP1* knockdown. Notably, the increase in *MMP13* gene expression in control cells (nt siRNA) after Sema3A stimulation was not detected on the protein level (Figure 5c). 

### 2.6. AKT Phosphorylation Is Impaired by Sema3A Stimulation in OA Chondrocytes

The AKT signaling pathway can be activated by phosphorylation within the carboxy- terminus at amino acids Ser473 or Thr308 of AKT1. Thereby, active AKT signaling can promote cell growth and cell survival by inhibiting apoptosis [12]. We observed a significant reduction of AKT-phosphorylation within 30 min after Sema3A treatment in control OA chondrocytes (nt siRNA), whereas in *NRP1* knockdown cells, the pAKT/total AKT ratio remains mostly unaffected during Sema3A stimulation (Figure 6a,b). 

In contrast, the phosphorylation status of ERK remained unaltered in both groups after Sema3A stimulation. The pERK/ERK ratio increased in both cell groups to the same extent during 60 min of Sema3A treatment (Appendix A). Hence, the phosphorylation of ERK is presumably not linked to the Sema3A/Nrp-1 signaling axis in OA chondrocytes. 

## 3. Discussion

We demonstrated in our study an increased expression of Sema3A in OA chondrocytes on the gene and protein level in comparison with that of non-OA chondrocytes. Sema3A is a diffusible axonal chemo-repellent factor that plays a critical role in the guidance of sensory nerve fibers and is thus involved in nervous innervation of different tissues [5]. Changes in peripheral joint innervation are supposed to be partly responsible for degenerative alterations in joint tissues, which contribute to the development of osteoarthritis (OA) [3]. Articular chondrocytes express Nrp1, the receptor for Sema3A, thus allowing response to Sema3A after binding, indicating that Sema3A may have also other functions besides that of a chemo-repellent substance. We hypothesized that Sema3A may act in an autocrine/paracrine manner, modulating chondrocyte metabolism in a catabolic way, supported by our findings that *MMP13* gene expression increases and AKT phosphorylation was compromised after Sema3A treatment. Our observation that expression of Sema3A is induced in OA chondrocytes was in line with the data from Okubo et al. [10]. However, in contrast to their results and the data from Sun et al. [13], the *NRP1* expression in our study was increased on the mRNA level only, but not on the protein level. This may be due to the mechanisms of Nrp1 receptor internalization, recycling and re-sensitization, also known to occur in other cell types, such as endothelial cells [14]. Using whole cell lysates does not reflect possible alterations in subcellular Nrp1 receptor localization, which may be the cause for a similar Nrp1 concentration in OA- and non-OA chondrocytes. Notably, Nrp1 needs a specific co-receptor, called Plexin A1, to transduce Sema3A signals within the cells [15]. The analysis of Plexin A1 expression was not part of our experimental design but might be a compensatory option for OA chondrocytes to modulate the Sema3A signaling activity that regulates *MMP13* gene expression among other effects.

During in vitro expansion of OA chondrocytes, Sema3A expression decreased gradually from passage 0 to passage 3. It is known that the culturing of chondrocytes in monolayer leads to dedifferentiation of the cells, thus suggesting a correlation between differentiation status and Sema3A expression level in OA chondrocytes. The phenotype and expression profile of articular chondrocytes changes during dedifferentiation, and the ability to produce type II collagen and aggrecan decreases and is eventually lost after approximately four passages. Instead, type I collagen and fibronectin are secreted and dominate the ECM macromolecular structure of the extracellular matrix (ECM) produced by dedifferentiated chondrocytes [16,17,18,19]. With this macromolecular change in ECM composition, human articular chondrocytes also lose their ability to produce a well-differentiated articular cartilage ECM and may enter a state of senescence in which they remain metabolically active but cease to proliferate [20]. Simultaneously with this process of dedifferentiation and decrease in Sema3A expression, the expression of Nrp1 receptor strongly increased from passage 0 to passage 3. This might be a direct cellular response to the reduction in Sema3A expression and an effort to counteract this. Thus, Nrp1 and Sema3A expression correlate reciprocally during passaging (aging) and relate to the (de-) differentiation status of chondrocytes.

The increased expression of Sema3A and Nrp1 in OA chondrocytes compared with “healthy” non-OA chondrocytes, prompted us to hypothesize that the Sema3A-Nrp1 signaling drives OA progression and acts in a catabolic manner on chondrocytes’ metabolism. To test this, we silenced the Nrp1 receptor via specific siRNA in OA chondrocytes and observed a catabolic change in metabolism, demonstrated by decreased proliferation and enhanced senescence. The inhibition of Nrp1 in chondrocytes might thereby help to prevent progression of dedifferentiation, as Nrp1 strongly increased during aging/passaging. A slow growth rate is a typical feature of well-differentiated chondrocytes [21], whereas proliferation of chondrocytes is induced again at an early stage of OA progression [22], indicating a chondro-protective role of Nrp1 inhibition in terms of preventing aberrant proliferation and dedifferentiation. In conclusion, we propose that there is a fine-tuned regulation of Nrp1 and Sema3A expression, responding to and reflecting the metabolic state and differentiation status of OA chondrocytes.

In addition to the increase in aging/senescence and the decrease in proliferation, we also observed an increase in the adhesion capacity of *NRP1* knockdown cells. More than 20 years ago, Nrp1 was first identified and characterized as an adhesion protein in the central nervous system (CNS), before it was also proven to be a receptor for Sema3A [23]. Cell contact between chondrocytes and their ECM is mainly mediated via integrins, which are matrix receptors critical for cell–matrix interactions [24]. α10β1 integrin, expressed by differentiated chondrocytes, preferably binds to collagen type II and collagen type IX and is a crucial focal adhesion contact (FAC) protein in chondrocytes [24]. Gene expression analysis after *NRP1* knockdown revealed no changes in *ITGA10*, nor in *COL1A1, COL2A1* or *COL3A1* expression, indicating the involvement of other integrins, i.e., one or two or other FAC components downstream of integrins in the adhesion process. We suspect that integrin activity, which was not analyzed in this study, is more meaningful for an integrin response, and may be modulated independently of the integrin gene expression. 

The most striking observation in gene expression analysis after *NRP1* inhibition was the strong decrease in MMP13 (~75%). MMP-13 is thought to be central to the irreversible degradation of the cartilage type II collagen fibrillar network in OA [25]. Our gene expression data obtained for *MMP13* suggest that the Sema3A-Nrp1 signaling may be critical for Mmp13-mediated cartilage degradation in OA. Of note, only two out of four cartilage samples from OA patients revealed a clear decrease in Mmp13 protein expression, whereas we have detected for all patients a reduced *MMP-13* gene expression level after *NRP1* knockdown. That observation indicates that there is a patient-specific modulating function of Nrp1 on Mmp13 protein expression. In addition, it is also critical to note here that the OA severity and status of the disease, the entire health situation of the patients and the medication may strongly vary within the OA group, contributing to interpatient differences. 

To verify whether the effect of Mmp13 reduction in *NRP1* knockdown cells is directly due to the inability of Sema3A to bind to the Nrp1 receptor, we stimulated control cells (nt siRNA = non targeting siRNA) and *NRP1* knockdown cells with Sema3A. In the control cells, (with an unaltered protein level of the Nrp1 receptor), a clear increase in *MMP13* gene expression was induced when stimulated with Sema3A. *NRP1* knockdown cells decreased *MMP13* gene expression after Sema3A treatment in comparison with non-Sema3A-stimulated *NRP1* knockdown cells, suggesting a direct impact of Sema3A-Nrp1 signaling on the level of transcriptional control of *MMP13* expression. The Mmp13 protein level was not altered in our experimental setting, which may be due to the time point of analysis or to other mechanisms of molecular regulation, such as endogenous inhibitors, proteases or non-coding RNAs [26]. 

Our data revealed that intact Sema3A-Nrp1 signaling is required for the transcription of the protease MMP13 in OA chondrocytes, presumably through impairing the pro-survival signaling pathway PI3K/AKT. Sun et al. support our data [13], showing that excessive Sema3A signaling activates chondrocyte apoptosis via inhibition of PI3K/AKT signaling. As the pERK/ERK ratio increased in *NRP1* knockdown and control cells to the same extend during 60 min of Sema3A treatment (Appendix A), we conclude that the ERK signaling pathway is presumably not linked to the Sema3A/Nrp1 signaling axis in OA chondrocytes.

## 4. Materials and Methods 

### 4.1. Isolation and Culture of OA- and Non-OA Chondrocytes 

All experiments were performed with human chondrocytes isolated from knee cartilage of OA patients who had undergone endoprosthetic surgery, and from healthy knee cartilage harvested from cadaver donors with no signs of OA (termed non-OA cartilage). We used cartilage tissue from 16 OA-patients (6 male and 10 female) and 10 non-OA donors (6 male and 4 female) in the study. The average age of the OA patients was 62 years (+/−9 years) and for the non-OA donors 30 years (+/−11 years). The use of human cartilage tissue was approved by the ethics committee at the University of Regensburg (Az: 14-101-0189, email: ethikkommission@klinik.ukr.de).

The cartilage was cut into small pieces and digested by collagenase type 2 (Worthington, Lakewood, CO, USA) for 16 h at 37 °C. The chondrocytes were then seeded at a density of 1.2 × 10^4^ cells per cm^2^ in T175 vials (Corning, Corning, NY, USA) and cultivated with 25 mL culture medium consisting of DMEM-F12 Ham (Sigma-Aldrich, St. Louis, MO, USA), 10% FCS (Sigma-Aldrich, St. Louis, MO, USA) and 1% Pen/Strep (Sigma-Aldrich, St. Louis, MO, USA) in an incubator (Thermo Fisher Scientific, Waltham, MA, USA) at 37 °C and 5% CO_2_. After 5 days, the culture medium was changed for the first time. The adherent chondrocytes, referred to as passage 0 at this point, were expanded up to a confluence of 70–80% by changing the medium twice a week, and then either split 1:5 for passaging, frozen in liquid nitrogen for storage or directly used for further experiments. 

### 4.2. RNA Isolation and Real-Time RT-PCR from OA- and Non-OA Chondrocytes

Total cellular RNA was isolated using the Absolutely RNA Miniprep Kit (Stratagene, San Diego, CA, USA) according to the manufacturer’s instructions. To generate single-stranded cDNA, RNA was reverse transcribed with an AffinityScript QPCR cDNA Synthesis Kit (Stratagene, San Diego, CA, USA) and PCR was performed with the Mx3005P QPCR System from Agilent Technologies using Brilliant II SYBER Green qPCR Mastermix (Agilent Technologies, Santa Clara, CA, USA) [26]. Gene expression of chondrocyte markers was analyzed relatively, calibrated to the expression in control cells (Calibrator), and normalized to GAPDH, TBP and 18s using following primer (Table 1): 

### 4.3. Protein Extraction and Western Blot Analysis

Total cell lysates were prepared with RIPA buffer (Invitrogen/Thermo Fisher, Waltham, MA, USA) containing proteinase inhibitor (Roche, Munich, Germany) and phosphatase inhibitor (Roche, Munich, Germany). Protein concentration was determined with the BCA assay (Invitrogen/Thermo Fisher, Waltham, MA, USA) and lysate aliquots containing 25–50 µg of total protein (depending on the protein of interest) were boiled for 5 min with SDS-sample buffer containing β-mercapto-ethanol (Merck, Darmstadt, Germany) and then subjected to a 10–15% SDS-PAGE. After electrophoretic separation, the proteins were transferred to nitrocellulose membranes (Bio-Rad, Hercules, CA, USA), blocked with 5% dried milk (Carl Roth, Karlsruhe, Germany), and subsequently incubated with primary antibodies for 16 h at 4 °C or 1 h at room temperature (Table 2): 

After washing, the membranes were incubated with rabbit HRP (horseradish peroxidase)-coupled secondary antibodies. Proteins were detected using ECL detection reagents (Invitrogen/Thermo Fisher, Waltham, MA, USA). Western blot signals were analyzed densitometrically using Photoshop CS3, and the ratio between the protein of interest and β-actin, or between the phosphorylated and non-phosphorylated proteins, was calculated.

### 4.4. Nrp1 siRNA Transfection

A pool of 5 different *NRP1* siRNAs was purchased from Dharmacon (Dharmacon, Lafayette, CO, USA) and transfection was performed according to the manufacturer’s instruction at a final concentration of 25 nM for the *NRP1* siRNA. A pool of 4 different non-targeting (nt) siRNAs was used as negative control. DharmaFECT siRNA Transfection Reagent (Dharmacon, Lafayette, CO, USA) was applied for the transfection procedure. At 24–48 h after transfection, the cells were harvested for RNA or protein extraction or trypsinized and re-seeded for functional assays.

### 4.5. BrdU Proliferation Assay 

The proliferation of OA chondrocytes after *NRP1* knockdown was determined using a BrdU assay (Carl Roch, Karlsruhe, Germany). A total of 3 × 10^3^ cells were cultured in 96-well plates for 2 days in growth media before the BrdU reagent was added according to manufacturer’s protocol. The absorbance was read at 450/690 nm with a Tecan ELISA reader (Tecan, Mannedorf, Switzerland). Results were calculated as percentage of control cells (nt siRNA).

### 4.6. Senescence-Associated (SA) β-Galactosidase Assay

For measuring senescence, an equal numbers of cells (7 × 10^4^) were washed with PBS and harvested. For the determination of cellular senescence, we used the SA-β-gal Activity Senescence Assay Kit (Cell Biolabs, San Diego, CA, USA) according to manufacturer’s instruction. Cells were lysed in lysis buffer with protease inhibitor for 5 min on ice, and after centrifugation of cell debris (500× *g*), 25 µL cell lysate and 25 µL assay buffer were mixed in one well of a black 96-well plate and incubated for 1 h at 37 °C in the dark. After incubation, 25 µL of the mixture was transferred to a new well and 100 µL of stop solution was added. Measurement of senescence-associated β-galactosidase (SA-β-gal) activity was performed at a peak absorbance of 360, which can be quantified using a microplate reader (Tecan, Männedorf, Switzerland).

### 4.7. Adhesion Assay

For measuring the adhesion of OA chondrocytes after *NRP1* knockdown, 1000 cells/well were seeded in a 96-well plate without additional coating. Culture medium and non-adhered cells were removed 30 min later. Cells were washed with PBS and fixed with 1% Glutaraldehyd (Merck, Darmstadt, Germany) for 30 min at room temperature. Crystal violet staining solution (in 0.02% aqua dest.) (Carl Roth, Karlsruhe, Germany) was added to the wells for 15 min. After removing of the staining solution, wells were washed with deionized H_2_O and crystal violet was extracted from the cells by adding 70% ethanol for 3 h. Plates were shaken gently at room temperature for 15 min and 100 µL of the extracted crystal violet were transferred to a flat-bottom 96-well plates. Absorbance was measured at 590 nm in a microplate reader (Tecan, Männedorf, Switzerland).

### 4.8. Caspase-3/7 Assay

For assaying apoptosis of OA chondrocytes after *NRP1* knockdown, equal numbers of cells (5000) were seeded in black 96-well plates in triplicates. Caspase-3/7 enzymatic activity was measured as an indicator of apoptosis using the Apo-ONE Homogeneous Caspase-3/7 assay (Promega, Fitchburg, WI, USA) according to the manufacturer’s instructions. A non-fluorescent caspase substrate (Z-DEVD-R110), added to the HTB94 cells, was thereby cleaved into fluorescent molecules with an emission maximum at 521 nm and measured in a microplate reader (Tecan, Männedorf, Switzerland).

### 4.9. Wound Healing (Migration) Assay

Wound healing (migration) assay was performed to study directional cell migration in vitro by creating a “wound” (gap) in a confluent chondrocyte cell layer, followed by capturing images at distinct time points at 72 and 144 h. OA chondrocytes after *NRP1* knockdown were seeded (1 × 10^4^) into Culture-Insert 2 Well (IBIDI, Gräfelfing, Germany). After 24 h, a cell-free gap (wound) was created in the cell layer by the removal of the insert, and cell migration can be visualized using a bright field microscope. Cells were washed with PBS and cultured from then on in the serum-free medium. Photos were taken at the beginning and after 24 h, and the closure of the gap (=wound) was determined via Photoshop CS (Adobe Inc., San Jose, CA, USA) and ImageJ. The pixel number of the cell free area of the gap was determined and used for analysis. 

### 4.10. Sema3A Stimulation

A total of 5 × 10^4^ OA chondrocytes were seeded in 6-well plates and cultured for 1–2 days in growth media. Recombinant Sema3A (Abcam, Cambridge, UK; ab233670) was added to the medium in a concentration of 1, 10 or 100 ng/mL for 24 h before measuring MMP13 gene and protein expression. Before determination of AKT, ERK, phospho-AKT and phosphor-ERK protein, the OA chondrocytes were treated with 100 ng Sema3A for 0, 3, 5, 15, 30 and 60 min before RIPA lysates were generated with the stimulated cells. 

### 4.11. Statistics

Statistical analysis was performed using Prism 6 (GraphPad Software Inc., San Diego, CA, USA). Results are presented as the means ± SD. Each assay was performed in replicates and repeated at least in 3 independent experiments. Two-tailed Mann–Whitney tests were used as standard nonparametric tests, determining whether the medians of the two groups (experimental group vs. control) were significantly different. Exact *p*-values were calculated. A value of *p* ≤ 0.05 was considered statistically significant. For the semi-quantitative analysis of Western blot intensity (or if specifically denoted), unpaired *t*-tests were used.

## 5. Conclusions and Summary

The reduced activation of the pro-survival AKT signaling pathway is accompanied by increased *MMP13* gene expression in control (nt siRNA) OA chondrocytes after Sema3A stimulation, thus we conclude a positive regulatory link to MMP13 by Sema3A signaling via inhibition of AKT activation. In line with that data, inhibition of Sema3A-Nrp1 signaling decreased *MMP13* gene expression with no alteration in AKT phosphorylation (Figure 7). In summary, we present strong evidence for a catabolic influence of the Sema3A-Nrp1 signaling axis in OA chondrocytes via compromising AKT phosphorylation and thus inhibiting the pro-survival influence of the AKT signaling pathway. We suggest that inhibiting the AKT signaling cascade may lead to an acceleration of cartilage degradation via increased Mmp13 expression. We hypothesize that targeting the Sema3A-Nrp1 signaling axis might present an opportunity to interfere with the progression of OA pathogenesis. 

## Figures and Tables

**Figure 1 ijms-23-14180-f001:**
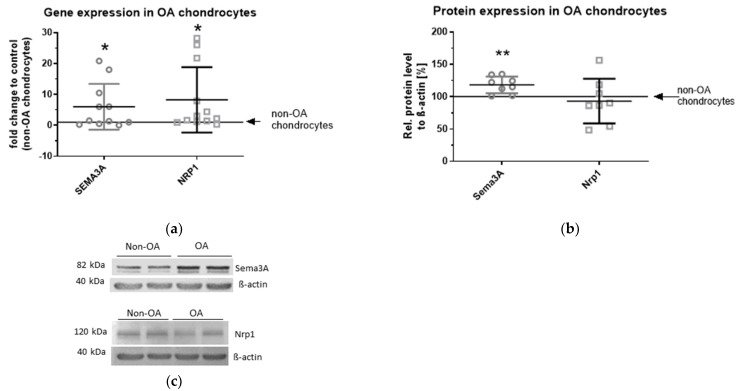
Expression of Sema3A and Nrp1 in chondrocytes from OA patients compared with non-OA donors. (**a**) Relative gene expression of *SEMA3A* and *NRP1* in OA chondrocytes in comparison with non-OA chondrocytes. n = 11–12; one sample *t*-test; * *p* ≤ 0.05; (**b**) Relative protein expression of Sema3A and Nrp1 was determined densitometrically. Expression of β-actin served as endogenous loading control. One sample *t*-test; n = 8, ** *p* ≤ 0.01; (**c**) Representative Western blot images displaying Sema3A, Nrp1 and β-actin protein expression in chondrocytes from OA and non-OA donors (2 different representative donors in each group are shown).

**Figure 2 ijms-23-14180-f002:**
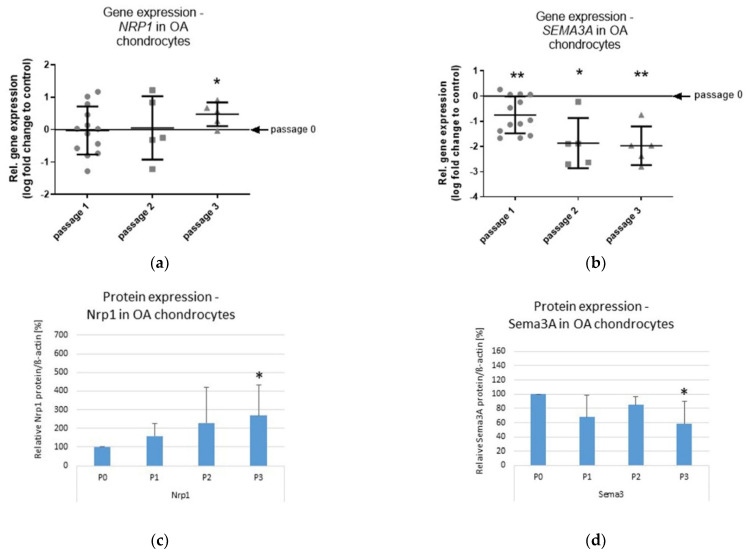
Influence of passage number on Sema3A and Nrp-1 expression in OA chondrocytes. (**a**) Relative *NRP1* gene expression in OA chondrocytes in passage 1, 2 and 3 in comparison with passage 0 (control). One sample *t*-test; n = 13 for passage 0 and 1, n = 5 for passage 2 and 3, * *p* ≤ 0.05; (**b**) Relative *SEMA3A* gene expression in OA chondrocytes in passage 1, 2 and 3 in comparison with passage 0 (control). One sample *t*-test; n = 5–13, * *p* ≤ 0.05; ** *p* ≤ 0.01; (**c**) Relative protein expression of Nrp1 was analyzed via Western blotting and determined densitometrically in passage 0 (P0), 1 (P1), 2 (P2) and 3 (P3). Expression of β-actin served as endogenous loading control; n = 4, * *p* ≤ 0.05; (**d**) Relative protein expression of Sema3A was analyzed via Western blotting and determined densitometrically in passage 0 (P0), 1 (P1), 2 (P2) and 3 (P3). Expression of β-actin served as endogenous loading control; n = 4, * *p* ≤ 0.05; (**e**) Representative Western blot image for Nrp1, Sema3A, and β-actin protein expression in chondrocytes from OA patients expanded in monolayer until passage 3.

**Figure 3 ijms-23-14180-f003:**
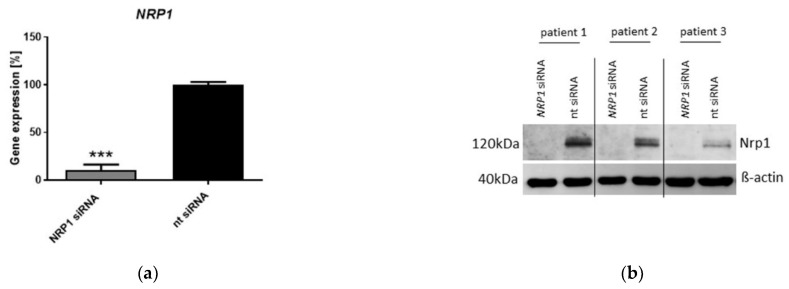
*NRP1* knockdown in OA chondrocytes. (**a**) Relative *NRP1* gene expression in OA chondrocytes after transfection with *NRP1* siRNA or nt (non-target) siRNA. One sample *t*-test; n = 8, *** *p* ≤ 0.001; (**b**) Representative Western blot pictures for Nrp1 and β-actin protein expression after transfection with *NRP1* siRNA or nt siRNA in chondrocytes of three different donors are shown. (**c**) Proliferation was determined after transfection with *NRP1* siRNA or nt siRNA. One sample *t*-test; n= 7, * *p* ≤ 0.05; (**d**) Adhesion was determined after transfection with *NRP1* siRNA or nt siRNA. One sample *t*-test; n = 7, ** *p* ≤ 0.01; (**e**) Senescence was determined after transfection with *NRP1* siRNA or nt siRNA. One sample *t*-test; n = 4, * *p* ≤ 0.05.

**Figure 4 ijms-23-14180-f004:**
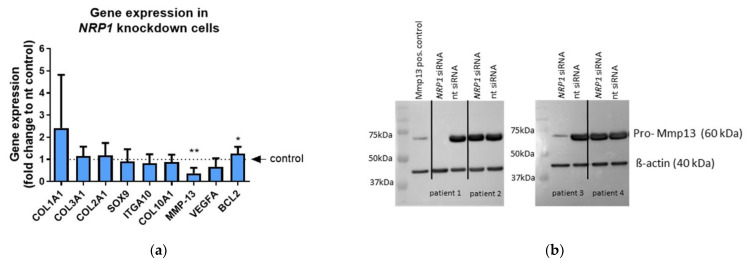
Gene expression of different marker genes and Mmp13 protein expression after *NRP1* knockdown in OA chondrocytes. (**a**) Relative gene expression analysis in *NRP1* knockdown cells in comparison with control (nt siRNA) cells. One sample *t*-test, n = 10, * *p* ≤ 0.05; ** *p* ≤ 0.01; (**b**) Western blot image of pro-Mmp13 and β-actin protein expression of different OA-patients (n = 4) after transfection with *NRP1* siRNA or nt siRNA.

**Figure 5 ijms-23-14180-f005:**
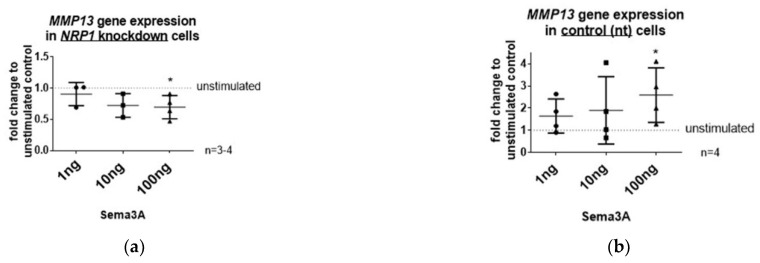
Sema3A stimulation of *NRP1* knockdown cells and control OA chondrocytes. (**a**) Relative gene expression of *MMP13* was determined in *NRP1* knockdown cells after stimulation with 1, 10 and 100ng recombinant Sema3A. Unstimulated *NRP1* knockdown cells served as control. One sample *t*-test, n = 3–4, * *p* ≤ 0.05; (**b**) Relative gene expression of *MMP13* was determined in control (nt) cells after stimulation with 1, 10 and 100 ng Sema3A. Unstimulated control (nt) cells served as control. One sample *t*-test, n = 3–4, * *p* ≤ 0.05; (**c**) Protein expression of Mmp13 was determined in *NRP1* knockdown and control cells after stimulation with 100 ng Sema3A. Representative Western blot is shown. (BioRender was used to create Figure 5a,b).

**Figure 6 ijms-23-14180-f006:**
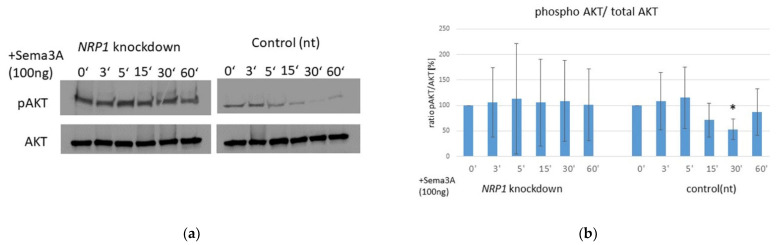
Sema3A stimulation leads to a decrease in AKT phosphorylation in OA chondrocytes. (**a**) Representative Western blot of phospho-AKT (pAKT) and total-AKT (AKT) protein after stimulation of *NRP1* knockdown and control cells (nt) with 100 ng/mL Sema3A for 0, 3, 5, 15, 30 and 60 min. (**b**) Relative protein expression of pAKT and AKT was determined densitometrically. The relative ratio of pAKT/total-AKT is shown; One sample *t*-test, n = 4, * *p* ≤ 0.05.

**Figure 7 ijms-23-14180-f007:**
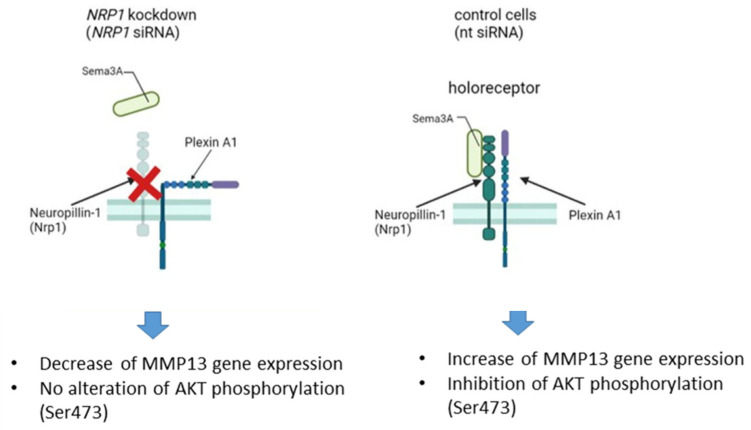
Schematic summary of Sema3A-Nrp1 signaling in OA chondrocytes. Inhibition of Sem3A-Nrp1 binding via NRP1 knockdown leads to a decrease in *MMP13* gene expression but no alteration in AKT phosphorylation. Control cells with active Sema3A-Nrp1 signaling axis revealed an increase in *MMP13* gene expression together with an impaired AKT phosphorylation.

**Table 1 ijms-23-14180-t001:** Primer sequences used in this study.

Gene	Forward	Reverse
*SEMA3A*	GGAGACTTGGTATGATTTAGAAGAGG	TGCTGAAATAGCAGTCGGTTC
*NRP1*	GCCACAGTGGAACAGGTGAT	CTATGACCGTGGGCTTTTCT
*COL1A1*	ACGTCCTGGTGAAGTTGGTC	ACCAGGGAAGCCTCTCTCTC
*COL3A1*	CTTCTCTCCAGCCGAGCTTC	TGTGTTTCGTGCAACCATCC
*COL2A1*	CCAGATGACCTTCCTACGCC	TTCAGGGCAGTGTACGTGAAC
*SOX9*	GTACCCGCACTTGCACAAC	TCTCGCTCTCGTTCAGAAGTC
*ITGA10*	GTGTGGATGCTTCATTCCAG	GCCATCCAAGACAATGACAA
*COL10A1*	CCC TCT TGT TAG TGC CAA CC	AGA TTC CAG TCC TTG GGT CA
*MMP13*	GACTGGTAATGGCATCAAGGGA	CACCGGCAAAAGCCACTTTA
*VEGFA*	CTTGCCTTGCTGCTCTAC	ACCACTTCGTGATGATTCTG
*BCL2*	ATGTGTGTGGAGAGCGTCAA	ACAGTTCCACAAAGGCATCC
*GAPDH*	CTGACTTCAACAGCGACACC	CCCTGTTGCTGTAGCCAAAT
*TBP*	GAACATCATGGATCAGAACAACA	ATAGGGATTCCGGGAGTCAT
*18s*	CTGGATACCGCAGCTAGGAA	GAATTTCACCTCTAGCGGCG
*TFRC*	ATC GGT TGG TGC CAC TGA AT	TTG CTG GTA CCA AGA ACC GC

**Table 2 ijms-23-14180-t002:** Primary and secondary antibodies used in this study.

Antibody	Company	Dilution
rabbit monoclonal anti-p44/42 MAPK (ERK1/2)	Cell Signaling Technology (CST), Danvers, MA, USA; #4695	1:2000
rabbit monoclonal anti-phospho-p44/42 MAPK (p-ERK1/2)	Cell Signaling Technology (CST), Danvers, MA, USA; #4370	1:1000
rabbit monoclonal anti-phospho-Akt (Ser473) (D9E) XP^®^	Cell Signaling Technology (CST), Danvers, MA, USA; #4060	1:2000
rabbit monoclonal anti-Akt (pan) (C67E7)	Cell Signaling Technology (CST), Danvers, MA, USA; #4691	1:1000
Semaphorin 3A	Abcam, Cambridge, UK; ab199475	1:1000
Neuropilin-1	Abcam, Cambridge, UK; ab81321	1:2000
MMP13	Abcam, Cambridge, UK; ab39012	1:3000
rabbit polyclonal anti-beta actin	Abcam, Cambridge, UK; #ab8227	1:5000
rabbit HRP (horseradish peroxidase)-coupled (secondary) antibody	Jackson Immuno Research, West Grove, PA, USA; #AB_2313567	1: 10,000

## Data Availability

Data are available on request due to privacy restrictions. The data presented in this study are available on request from the corresponding author.

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
