# Peer review of "Semaphorin 3A-Neuropilin-1 Signaling Modulates MMP13 Expression in Human Osteoarthritic Chondrocytes"

_ijms, 2022, doi:10.3390/ijms232214180_

Round 1

Reviewer 1 Report

The authors examined the importance of chemo –repellent for sensory nerve gene Sema3A and its receptor NRP1 for OA chondrocyte viability, proliferation, adhesion, senescence, apoptosis, migration, and expression of chondrocyte specific genes. They found that Sema3A-NRP1 signaling inhibits pro-survival AKT signaling capable of upregulating MMP-13 gene expression.

Comments

1.      Title should be rephrased: it is necessary to indicate how exactly Sema3A-NRP1 signaling affects MMP-13 expression.

2.      Line 13-15: The authors should indicate the type of chondrocytes used for comparison of the expression of Sema3A and NRP1 genes in OA chondrocytes.

3.      Fig. 1a):  It is not clear why the number of dots on the graph does not correspond to n=15. This should be clarified.

4.      Fig. 2a) and b): It is not clear why number of samples decreased at passage 2 and 3 compared to passage 1. This should be clarified.

5.      Line 100: n=5-15. This should be clarified.

6.      Fig. 7 should be removed as it reproduced Fig.5.

7.      Line 249-250: The authors cannot make conclusion on the changes of authors cannot make conclusion on the changes of integrin activity as this issue was not examined. This should be corrected.

8.      Lines 363: The incubation time with crystal violet is not clear. This should be clarified.

9.      Lines 379-380: The protocol for wound healing (migration) assay should be described in more detail. This should be corrected.

10.  Overall: The rationale for chemo –repellents for sensory nerve genes in the development and progression of OA should be discussed. 

Author Response

We sincerely thank both reviewers for their constructive and helpful comments, which we have addressed point by point.

Please note that the file called "author-coverletter-23923440.v3" is the revised manuscript ijms-2026273. Unfortunately, the system has renamed the manuscript incorrectly!

Reviewer 1:

Comments and Suggestions for Authors

The authors examined the importance of chemo –repellent for sensory nerve gene Sema3A and its receptor NRP1 for OA chondrocyte viability, proliferation, adhesion, senescence, apoptosis, migration, and expression of chondrocyte specific genes. They found that Sema3A-NRP1 signaling inhibits pro-survival AKT signaling capable of upregulating MMP-13 gene expression.

Comments

  1. Title should be rephrased: it is necessary to indicate how exactly Sema3A-NRP1 signaling affects MMP-13 expression.

Response: We suggest: “Semaphorin 3A-Neuropilin-1 signaling modulates MMP13 expression in human osteoarthritic chondrocytes”

  1. Line 13-15: The authors should indicate the type of chondrocytes used for comparison of the expression of Sema3A and NRP1 genes in OA chondrocytes.

Response: We have added following text “compared to non-OA control chondrocytes (line 15)” in the abstract.

  1. Fig. 1a): It is not clear why the number of dots on the graph does not correspond to n=15. This should be clarified.

Response: Thank you for the comment. This error has been corrected to “n=11-12” in the legend of figure 1a.

  1. Fig. 2a) and b): It is not clear why number of samples decreased at passage 2 and 3 compared to passage 1. This should be clarified.

Response: When starting to work on the study we analyzed the passage-dependent expression of Sema3A and NRP-1 with chondrocytes obtained from 13 different patients meaning N=13 until passage 1. However, further expansion of the cells to passage 2 and 3 was then only done exemplarily with N=5 of the original N= 13 chondrocyte samples due to time and material reasons.

  1. Line 100: n=5-15. This should be clarified.

Response: This statement has been revised in the figure legend of fig. 2 to “n=13 for passage 0 and 1, n=5 for passage 2 and 3”.

  1. Fig. 7 should be removed as it reproduced Fig.5

Response: We have removed the schematic receptor illustration in Figure 5 but left it in Figure 7, which is intended as a summarizing figure.

  1. Line 249-250: The authors cannot make conclusion on the changes of integrin activity as this issue was not examined. This should be corrected

Response: Thank you for this point. Our statement was not meant to be a conclusion about integrin activity. Rather, it is a possible explanation as we did not find any change in integrin gene expression (ITGA10) in our studies.

Sorry for this ambiguous expression. We have rewritten the sentence to:

We suspect that integrin activity, which was not analysed in this study, is more meaningful for an integrin response, and may be modulated independently of the integrin gene expression.

  1. Lines 363: The incubation time with crystal violet is not clear. This should be clarified.

Response:  Incubation time of crystal violet for the adhesion assay is indicated as 15 min in line 370.

  1. Lines 379-380: The protocol for wound healing (migration) assay should be described in more detail. This should be corrected.

Response: Thank you for your comment. The protocol has been explained in more detail under “4.9. Wound healing (migration) assay” starting at line 384.

  1. Overall: The rationale for chemo –repellents for sensory nerve genes in the development and progression of OA should be discussed.

Response: Thank you for that advice. We discussed that point in more detail in the discussion section (line 194 to 200).

“Sema3A is a diffusible axonal chemo-repellent factor that plays a critical role in the guidance of sensory nerve fibers and is thus involved in nervous innervation of different tis-sues [5]. Changes in peripheral joint innervation are supposed to be partly responsible for degenerative alterations in joint tissues, which contribute to the development of osteoarthritis [3]. Articular chondrocytes express Nrp-1, the receptor for Sema3A, thus, allowing to respond to Sema3A after binding, indicating that Sema3A may have also other functions besides that of a chemo-repellent substance.”

Reviewer 2 Report

The paper seeks to investigate the changes in expression of Sema3a and its receptor Nrp1 in chondrocytes isolated from patients with OA.

Overall the experiments have been performed well but are largely simplistic in nature with regard to provision of mechanistic evidence.

In section 2.3 the results describe the knockdown of Nrp1 and investigation of the effect on function. Whilst physiology and biology are investigated there are no experiments that relate to the function of the cells.

Growth rate of cells is also referenced; however, this implies cell size is increasing, better to use more appropriate terminology such as the proliferation of the cells.

The investigators also refer to investigating the metabolism of cells, and yet cell metabolism is not studied.

The figures are very simplistic and poorly presented. The graphs for instance have very little resolution on axis.

Author Response

We sincerely thank both reviewers for their constructive and helpful comments, which we have addressed point by point.

Please note that the file called "author-coverletter-23923440.v1" is the revised manuscript ijms-2026273. Unfortunately, the system has renamed the manuscript incorrectly!

Reviewer 2:

Comments and Suggestions for Authors

The paper seeks to investigate the changes in expression of Sema3a and its receptor Nrp1 in chondrocytes isolated from patients with OA.

Overall the experiments have been performed well but are largely simplistic in nature with regard to provision of mechanistic evidence.

Response: Here, we fully agree with the reviewers’ opinion.

  1. In section 2.3 the results describe the knockdown of Nrp1 and investigation of the effect on function. Whilst physiology and biology are investigated there are no experiments that relate to the function of the cells.

Response:  Thank you for that comment and important hint. We changed in line 113/114 “functional aspects” to “cell biology” to choose a more precise expression. Since the function of NRP-1 in chondrocytes in the context of osteoarthritis has not been studied in detail up to now, we cannot draw any conclusions about the function of the NRP1- knockdown cells, but can only describe the changes in cell physiology and biology and thus provide the basis for the elucidation of the function.

  1. Growth rate of cells is also referenced; however, this implies cell size is increasing, better to use more appropriate terminology such as the proliferation of the cells.

Response: Thank you for that comment. We have replaced growth rate by “proliferation” in line 115

  1. The investigators also refer to investigating the metabolism of cells, and yet cell metabolism is not studied.

Response: Thank you for that helpful annotation. We consider cellular metabolism as the sum of all biochemical changes that take place in a cell through which energy and basic components are provided for essential metabolic processes, including the synthesis of new molecules and the breakdown and removal of others. All functional assays that we have performed in our study require energy. For example, we consider migration as an energy-intensive, multi-step process involving formation of adhesion plaques, cell adhesion to the substrate, formation of cell protrusions, and finally detachment. Each of these steps require cells to generate and consume energy, regulating their morphological changes and force generation. Therefore, we refer to investigating the metabolism of cells.

Sorry, if the wording was ambiguous here.

  1. The figures are very simplistic and poorly presented. The graphs for instance have very little resolution on axis.

Response: We are sorry but we have prepared all of the figures in the required resolution of 300 dpi. Perhaps, the journal editing team has an explanation why the resolution appears low. The system did not allow to upload the individual figures but only embedded in one word or PDF file.
